# Retrospective Genotyping of Enteroviruses Using a Diagnostic Nanopore Sequencing Workflow

**DOI:** 10.3390/pathogens13050390

**Published:** 2024-05-08

**Authors:** Vanessa van Ackeren, Stefan Schmutz, Ian Pichler, Gabriela Ziltener, Maryam Zaheri, Verena Kufner, Michael Huber

**Affiliations:** Institute of Medical Virology, University of Zurich, 8057 Zurich, Switzerland; vanackeren@imcr.uzh.ch (V.v.A.); schmutz.stefan@virology.uzh.ch (S.S.); pichler.ian@virology.uzh.ch (I.P.); ziltener.gabriela@virology.uzh.ch (G.Z.); zaheri.maryam@virology.uzh.ch (M.Z.); kufner.verena@virology.uzh.ch (V.K.)

**Keywords:** human pathogens, enterovirus, next-generation sequencing (NGS), novel diagnostic methods, outbreaks, surveillance

## Abstract

Enteroviruses are among the most common viruses pathogenic to humans. They are associated with various forms of disease, ranging from mild respiratory illness to severe neurological diseases. In recent years, an increasing number of isolated cases of children developing meningitis or encephalitis as a result of enterovirus infection have been reported, as well as discrete enterovirus D68 outbreaks in North America in 2014 and 2016. We developed an assay to rapidly genotype enteroviruses by sequencing a region within the VP1 gene using nanopore Flongles. We retrospectively analyzed enterovirus-/rhinovirus-positive clinical samples from the Zurich, Switzerland area mainly collected during two seasons in 2019/2020 and 2021/2022. Respiratory, cerebrospinal fluid, and stool samples were analyzed. Whole-genome sequencing was performed on samples with ambiguous genotyping results and enterovirus D68-positive samples. Out of 255 isolates, a total of 95 different genotypes were found. A difference in the prevalence of enterovirus and rhinovirus infections was observed for both sample type and age group. In particular, children aged 0–4 years showed a higher frequency of enterovirus infections. Comparing the respiratory seasons, a higher prevalence was found, especially for enterovirus A and rhinovirus A after the SARS-CoV-2 pandemic. The enterovirus genotyping workflow provides a rapid diagnostic tool for individual analysis and continuous enterovirus surveillance.

## 1. Introduction

Enteroviruses, along with rhinoviruses, influenza viruses, and coronaviruses, are among the most common respiratory pathogens [1,2]. The genus *Enterovirus* belongs to the order *Picornavirales* and the family *Picornaviridae* [3]. Enteroviruses are single-stranded, positively oriented RNA viruses ((+)-ss-RNA). They are distributed worldwide, with children being at higher risk for infection than adults for all species [4,5]. In Europe, *Enterovirus B* is the most prevalent species, followed by *Enterovirus A*, while *Enterovirus C* and *D* are less frequently detected. The types most often found are echovirus 30 and coxsackievirus A6, both of which belong to the species *Enterovirus B* [6,7]. In the last two decades, several large outbreaks of different enterovirus species have been reported. These include outbreaks of enterovirus D68 in North America, associated with acute flaccid myelitis (AFM) and encephalitis in 2014 and 2016 [8,9]. Similarly, Europe has seen an increase in severe cases of AFM since 2016 coinciding with enterovirus D68 positivity [10,11,12,13,14], while Asia and the Pacific have experienced major outbreaks of enterovirus A71 causing hand, foot, and mouth disease [6,7,10]. Due to the higher number of severe cases, the interest in enterovirus surveillance and epidemiological studies has increased dramatically [6,12,15,16,17,18,19,20,21,22,23].

The present study was designed to establish a fast and cost-effective amplicon genotyping workflow using nanopore sequencing and perform retrospective enterovirus genotyping before and during the SARS-CoV-2 pandemic.

## 2. Materials and Methods

### 2.1. Patient Samples

Residual routine patient samples with previously confirmed enterovirus infection from the archive of the Institute of Medical Virology, University of Zurich, were used. Enterovirus infections were qualitatively detected as part of routine diagnostic testing using the ePlex Respiratory Pathogen Panel (GenMark Diagnostics, Carlsbad, CA, USA), a Fast-Track Diagnostic Respiratory Pathogens 21 Kit (Fast-Track Diagnostic, Luxembourg), or BioFire Respiratory or Meningitis Panels (BioMérieux, Craponne, France).

Respiratory samples were randomly selected from two defined seasons with uniform distribution by month. If multiple samples were selected from the same patient within one season, only the first sample was included. Additionally, all stool and cerebrospinal fluid (CSF) samples were included throughout the complete study duration. All samples were anonymized irreversibly.

### 2.2. Nucleic Acid Extraction

Nucleic acids were extracted on the NucliSENS eMAG system (BioMérieux) according to the manufacturer’s instructions. For this procedure, 500 µL of each sample was eluted in 50 μL.

### 2.3. Enterovirus qPCR

Qualitative RT-qPCR was performed on all samples for pan-enteroviruses [24] and rhinoviruses (modified from [25]) using a TaqMan RT-PCR Mix and Ag-Path-ID^TM^ One-Step RT PCR chemistry (Applied Biosystems/Thermo Fisher Scientific, Waltham, MA, USA).

### 2.4. Enterovirus Amplicon Sequencing

For enterovirus genotyping, a region within the VP1 gene was amplified as described by Nix et al. [26]. First, 0.5 mM dNTPs and 0.1 µM of each primer AN32, AN33, AN34, and AN35 were added to 5 µL template in a total reaction volume of 13 µL and incubated at 65 °C for 5 min to denature secondary RNA structures. cDNA was generated using the SuperScript IV Reverse Transcriptase (Invitrogen/Thermo Fisher Scientific) as described by the manufacturer. Then, three consecutive PCRs were performed as described by Nix et al. and the WHO enterovirus surveillance guide [26,27] (Figure 1).

The first PCR produced an amplicon of about 800 bp. In a vial of 25 µL total volume, a reaction mix was prepared that contained 0.05 U/µL AmpliTaq Polymerase and 1× PCR Buffer (both Applied Biosystems/Thermo Fisher Scientific); 0.2 mM dNTPs; 1 µM of each forward and reverse primer SO224 and SO222, respectively; and 5 µL cDNA [24]. The cycling conditions were 95 °C for 2 min, 35 cycles of 95 °C for 15 s, 42 °C for 30 s, 72 °C for 45 s, and 72 °C for 5 min. The ramp rate between annealing and extension was restricted to 10%.

The second, semi-nested PCR produced an amplicon of about 400 bp. In a vial of 50 µL total volume, a reaction mix was prepared containing 0.05 U/µL AmpliTaq Polymerase; 1× PCR Buffer; 0.8 µM of each forward and reverse primer AN89 and AN88, modified to contain a nanopore adapter at the 5′-ends, respectively; and 1 µL product of the first PCR [23]. The cycling conditions were 95 °C for 2 min, 38 cycles of 95 °C for 15 s, 60 °C for 30 s, 72 °C for 45 s, and 72 °C for 5 min.

The third PCR with tailed primers produced an amplicon of about 430 bp. In a vial of 50 µL total volume, a reaction mix was prepared containing 1× LongAmp Hot Start Taq Master Mix (New England Biolabs, Ipswich, MA, USA); 0.2 µM barcoded primers (BP, i.e., BP01 to BP12 from the PCR Barcoding Kit SQK-PBK004, Oxford Nanopore Technologies (ONT, Oxford, UK); and 24 µL product of the second PCR diluted to 0.4 ng/µL (~0.2 ng/uL final concentration). The cycling conditions were 95 °C for 3 min, 14 cycles of 95 °C for 15 s, 56 °C for 15 s, 65 °C for 50 s, and 65 °C for 6 min.

The final amplicon was purified with 80% volume Agencourt AMPure XP beads (Beckman Coulter, Brea, CA, USA), washed twice with 70% ethanol, and eluted in 10 µL UltraPure DNase/RNase-free distilled water (Thermo Fisher Scientific).

Up to 12 samples were pooled in 5 µL and prepared to be sequenced on a flow cell dongle (Flongle, R9.4.1 chemistry, ONT) as described by the manufacturer’s instructions. Samples were sequenced for at least 4 h on a GridION X5 Mk1 (ONT). Reads were basecalled using guppy (v6.3.9 onwards, https://community.nanoporetech.com accessed on 26 April 2024).

### 2.5. Bioinformatic Analysis

Reads were analyzed with an in-house developed script using *Snakemake* (https://github.com/medvir/ONT_amplicon/releases/tag/v1.0 accessed on 26 April 2024). Briefly, reads from multiple FASTQ files per sample were combined, and their barcodes were trimmed before mapping using *minimap2* against user-provided reference sequences of all enterovirus species (538 sequences, https://github.com/medvir/ONT_amplicon/tree/main/references accessed on 26 April 2024). Incomplete or low-coverage consensus sequences were excluded from the final analysis. Genotyping was defined as successful if it resulted in an unambiguous genotyping call.

### 2.6. Untargeted Whole-Genome Sequencing

Whole-genome sequencing was performed according to a previously published metagenomic sequencing workflow [28,29]. Briefly, reverse transcription with random hexamers and second strand synthesis was performed prior to the construction of sequencing libraries using the NexteraXT protocol (Illumina, San Diego, CA, USA) and sequencing on a MiSeq for 1 × 151 cycles using version 3 chemistry. Bioinformatic analysis was performed using the pipeline VirMet (https://github.com/medvir/VirMet/releases/tag/v1.1.1 accessed on 26 April 2024), and for consensus sequence generation, SmaltAlign (https://github.com/medvir/SmaltAlign/releases/tag/v1.1.0 accessed on 26 April 2024) was used. Enterovirus D68 sub-genogroupes were determined using the enterovirus genotyping tool *RIVM* [30].

### 2.7. Estimation of Genotype Diversity

To estimate the genotype diversity, a rarefaction curve with an asymptotic approximation was created according to Zou et al. [31,32]. By plotting 100 collector curves, the average distribution curve was derived, and the maximum number of genotypes was estimated using the Michaelis–Menten equation [33].

## 3. Results

### 3.1. Sample Characteristics

Enteroviruses are present throughout the year and can cause infections even in the summer months; however, infections are seasonal with an increase in tested and positive samples from October to April. No such increase was observed during the SARS-CoV-2 pandemic in 2020–2021, and from March 2020 to March 2021, in particular, the total number of samples decreased (Figure 2).

Respiratory samples from two seasons, one before the onset of the pandemic (September 2019–March 2020) and another during the SARS-CoV-2 pandemic (July 2021–April 2022), were selected. Samples were randomly chosen for a retrospective enterovirus genotyping study (Appendix A), and 255 samples were included in the study.

Patient age distribution ranged from 0 to 89 years. The highest proportion of patients were children aged 0–4 years (26.7%). There was a gap in the 10–14-year age group, with no positive samples (Figure 3).

Respiratory samples accounted for the largest proportion (77.3%), followed by stool samples (14.1%) and CSF samples (6.3%). The median age of patients with respiratory samples was 52 years, whereas the median age of patients with stool samples was 0 years.

### 3.2. Targeted Amplicon Sequencing and Genotyping

In total, 255 samples were sequenced, while 36 initially yielded an ambiguous result. In total, 18 samples were resolved by whole-genome sequencing. Ultimately, 237 isolates (92.9%) were successfully genotyped. Three samples with a double infection of two different genotypes were found (Appendix A, study numbers 188, 266, and 406). Of those, two samples were infected with two genotypes from the same species, and one sample contained two genotypes from two different species.

Figure 4 shows the identified species, categorized into two age groups. In patients younger than five years, a predominance of enteroviruses A and B was observed, mainly detected in stool samples (*n* = 32) and CSF (*n* = 8), while rhinoviruses were detected to a lesser extent in this age group. In patients older than four years, rhinoviruses A–C were predominantly found in respiratory specimens (*n* = 155), whereas enteroviruses were detected less frequently in this age group. Notably, three cases of enterovirus D infection were detected, and no isolates belonging to enteroviruses C were found in either age group.

Overall, enterovirus B isolates were mostly detected in stool samples of children younger than five years (68.3%), whereas rhinovirus A isolates were most common in respiratory material from patients older than four years of age (54%).

Without distinguishing between age groups, enterovirus B was predominant in both stool samples (65.7%) and CSF samples (81.3%). Rhinovirus A was the most common virus found in respiratory samples (55.4%, Figure 5). 

Samples were classified on both species and genotype levels, which revealed a high variability of genotypes belonging to the species *Enterovirus A* and *B* and *Rhinovirus A*–*C* (Table 1). *Rhinovirus A* showed the highest variation with 41 different genotypes found. *Rhinovirus B* was represented with 17 different genotypes and *Rhinovirus C* with 18. Among the enteroviruses, the species *Enterovirus B* was found most frequently with 14 detected genotypes, *Enterovirus A* was represented with 4 detected genotypes, and for *Enterovirus D*, only genotype D68 was found. In total, 95 different genotypes for all species were detected. 

In addition, the analysis of genotypes by sample type (CSF, stool, and respiratory samples) revealed no distinct prevalence of any genotype across these materials.

### 3.3. Untargeted Whole-Genome Sequencing

Three types of samples were selected for further investigation using unbiased whole-genome sequencing instead of the established target-specific amplicon approach. These included (i) twelve samples with inwardly shifted primer binding sites, (ii) four samples with high positive results in both qPCR systems and ambiguous genotyping results, and (iii) three enterovirus D68-positive samples. Whole-genome sequencing was achieved for all 18 samples (Appendix A), and subsequent genotyping was successful for 18 of them.

One sample with a shifted primer binding site could not be classified at the genotype level. The whole-genome consensus sequence was aligned with BLAST and showed a 97% match with a sequence published by Tirosh et al. (MH899592.1) of a previously unclassified rhinovirus B [35] (Appendix A, study number 32). 

The whole-genome analysis of the enterovirus D68 samples allowed for subclassification into two clade A2 and one clade B3 enterovirus D68.

One co-infection with norovirus GII was detected in an enterovirus D68-positive CSF sample. In another sample, a co-infection with bocavirus was found (Appendix A, study numbers 116 and 273, respectively).

### 3.4. Influence of the SARS-CoV-2 Pandemic on Circulating Enterovirus Species

To answer the question of whether the SARS-CoV-2 pandemic had an impact on enterovirus distribution in respiratory samples, one season before and another during the SARS-CoV-2 pandemic were compared. In the respiratory season before the pandemic (2019/2020), *Rhinovirus A* accounted for 47.4% of the detected species, followed by a high proportion of *Rhinovirus B* (32.0%). *Rhinovirus C* was detected in 17.5% of the cases (Figure 6, left donut plot). By comparison, in the season 2021/2022, the proportion of *Rhinovirus A* has increased to 64.4%. The second most common species found in this season was *Rhinovirus C* (12.6%) (Figure 6, right donut plot). *Enterovirus A* was the third most common species (9.2%), which was not detected in the previous season. Overall, the proportion of *Enteroviruses A-D* increased from 3.1% in 2019/2020 to 14.9% in 2021/2022. The difference in the distribution of enterovirus species between the two seasons was significant (Fisher’s exact test with *p*-value < 0.0001). 

### 3.5. Estimation of Genotype Diversity 

To estimate the genotype diversity, a rarefaction analysis was performed (Figure 7). 

By generating multiple collector curves, i.e., species accumulation curves, the average distribution or rarefaction curve was calculated. The extrapolation of the curve approached the estimated number of genotypes [31,32]. Around 156 of the more than 200 known genotypes were estimated to be circulating in the Zurich area.

## 4. Discussion

Enteroviruses are a very diverse and variable genus, characterized by worldwide distribution. Research on enteroviruses has continued to grow, particularly due to regional outbreaks over the last 20 years [6,12,15,16,17,18,19,20,21,22,23]. Virus genotyping is likely to be of increasing interest in the coming years, particularly because of the increase in viral re-emergence due to continued population growth, urbanization, and increased global travel [36]. In this study, we developed a rapid diagnostic tool for enterovirus genotyping based on ONT sequencing for individual analysis and continuous enterovirus surveillance. Amplicon sequencing with ONT allows for the analysis of multiple samples per run and provides a cost-effective way to genotype enteroviruses. Compared to other workflows using Illumina sequencing, ONT provides an easier library preparation and faster time to result, due to real-time sequencing.

Its applicability was demonstrated by retrospectively genotyping 255 patient samples collected from 2019 to 2022 in the Zurich area. Overall, genotyping was successful in 92.9% of the analyzed cases, demonstrating the robustness of the established method. Even in 14 samples where both qPCR systems failed, the isolate could be genotyped, probably due to the different primer systems used for sequencing and qPCR.

The main finding was a high diversity of circulating enterovirus genotypes, with a total of 95 genotypes detected and an estimation of about 156 genotypes in circulation in the Zurich region. This high enterovirus diversity has already been shown in different studies [6,10,12]. Those studies refer specifically to samples from young children and are designed to identify enteroviruses A-D only [6,10,37]. In our study, we not only focused on enterovirus A-D species but also included rhinoviruses A-C. All in all, and in contrast to other studies, no predominance of individual genotypes was found.

We did not detect any enterovirus C species despite its reported presence in Europe [6,10]. The fact that enterovirus C infections are often associated with the gastrointestinal tract and that only 14.1% of the included samples were stool samples may explain why no enterovirus C was detected in this study.

Enteroviruses of species A and B were mainly found in stool and CSF samples, whereas rhinoviruses A–C were mainly found in respiratory material. The results underline that the frequency of species varies greatly in different patient materials. This observation has also been made in other studies [6,10]. In particular, rhinoviruses are mainly associated with respiratory infections and are therefore predominately found in respiratory material. Notably, respiratory samples were analyzed five times more than CSF and stool samples, which limits a direct comparison of genotype composition [6,12]. No specific genotype prevalence was observed in any material.

The prevalence of genotypes before and during the SARS-CoV-2 pandemic was also of interest. Enterovirus A was only found during the second respiratory season, and at the same time, rhinovirus A was more predominant. Whether this is a consequence of the hygienic measures taken during the pandemic is unknown. A recent study has shown continuing rhinovirus infection with 60% thereof being rhinovirus A, despite hygienic measures during the pandemic [38]. The reduction in the total number of positive enterovirus infections in 2020, which was also found by Fisher et al. [15], could have been a consequence of directing most diagnostic efforts toward SARS-CoV-2 and could have biased genotype detection toward more pathogenic enteroviruses. For a better understanding of genotype composition before and during the SARS-CoV-2 pandemic, a higher number of samples from more seasons would be needed.

In conclusion, this study highlighted the high diversity of genotypes observed both within and across respiratory seasons but did not reveal any enterovirus genotype(s) of significant abundance. 

## Figures and Tables

**Figure 1 pathogens-13-00390-f001:**
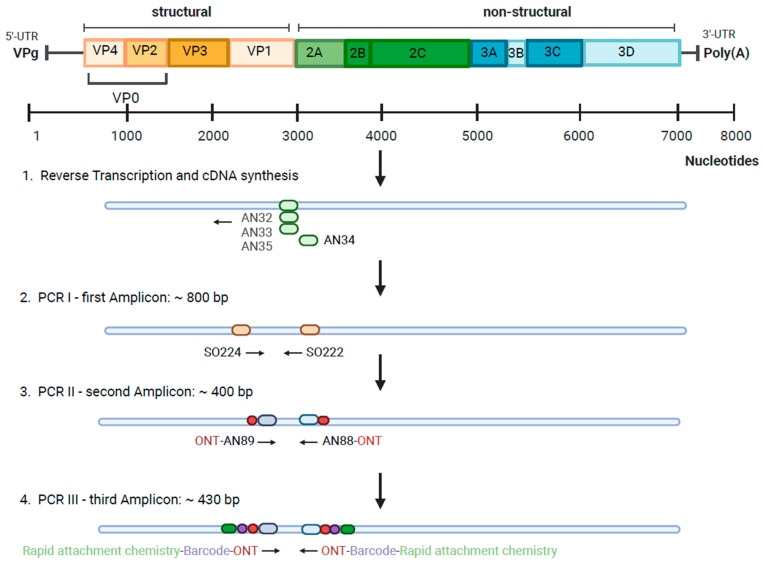
Schematic of enterovirus genome and amplification steps with primer binding sites for enterovirus genotyping (created with biorender.com).

**Figure 2 pathogens-13-00390-f002:**
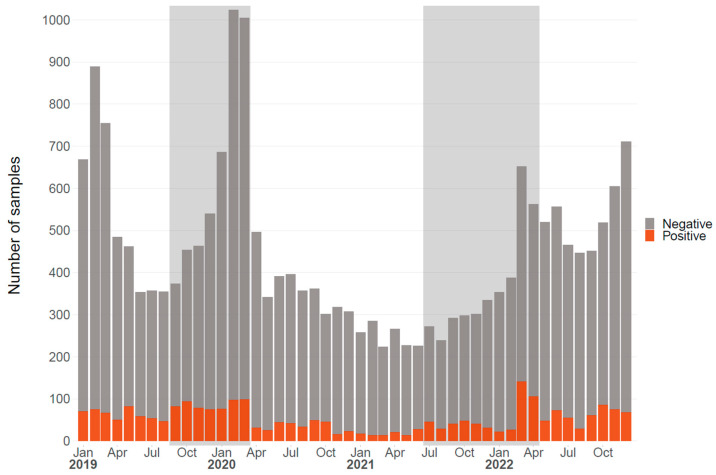
Number of samples tested for enterovirus at the Institute of Medical Virology, University of Zurich. Negative samples are indicated in gray, and positive samples are shown in red. The absolute numbers are stacked. The light gray areas show the two selected outbreak seasons.

**Figure 3 pathogens-13-00390-f003:**
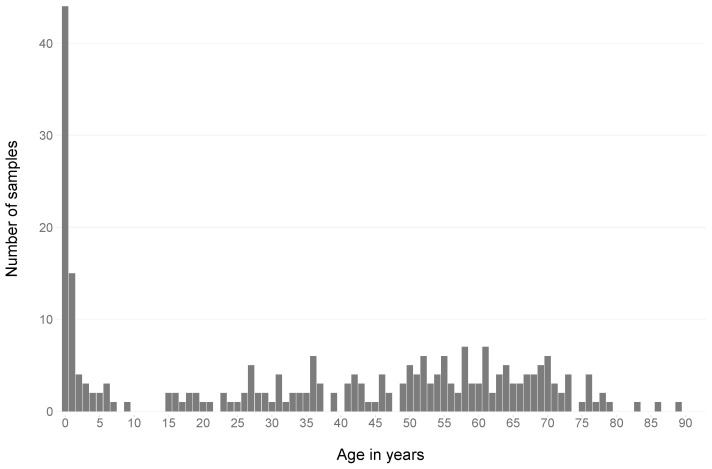
Age distribution of all patients from the 255 included isolates between July 2019 and December 2022.

**Figure 4 pathogens-13-00390-f004:**
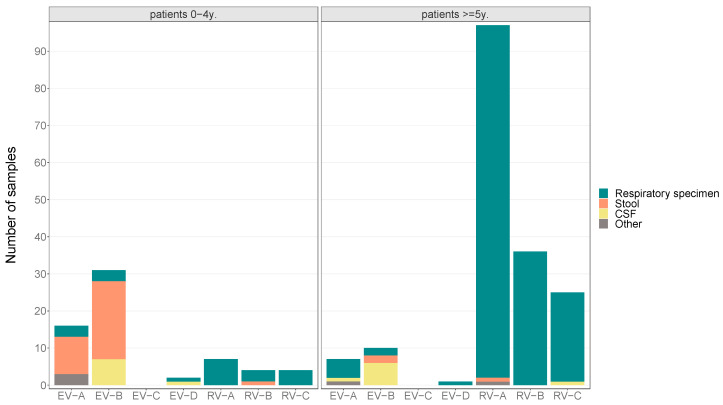
Number of enterovirus species among patients under and over five years of age by sample types.

**Figure 5 pathogens-13-00390-f005:**
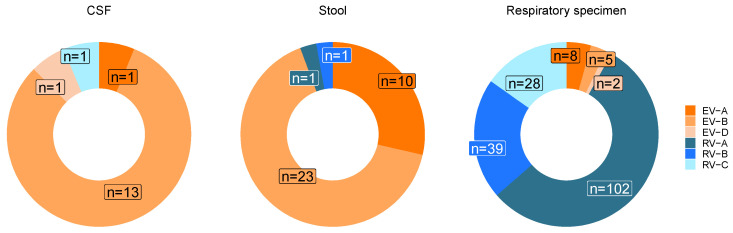
Distribution of enterovirus species in cerebrospinal fluid (CSF), stool samples, and respiratory material.

**Figure 6 pathogens-13-00390-f006:**
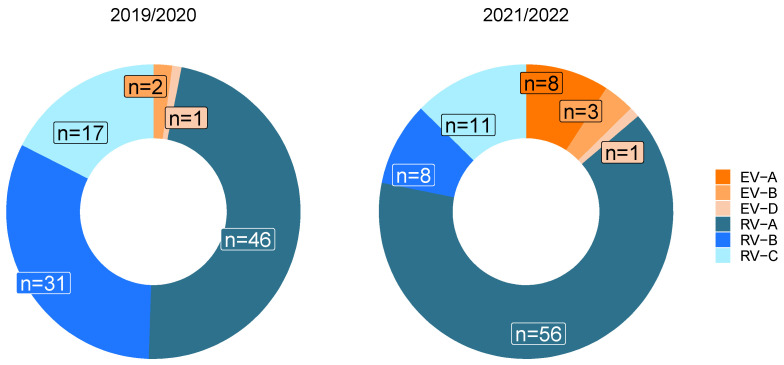
Distribution of detected enterovirus species in respiratory samples in the two defined outbreak seasons: the first from September 2019 to the end of March 2020, before the onset of the SARS-CoV-2 pandemic and the second from July 2021 to the end of April 2022 during the SARS-CoV-2 pandemic. Samples from all parts of the respiratory tract were used.

**Figure 7 pathogens-13-00390-f007:**
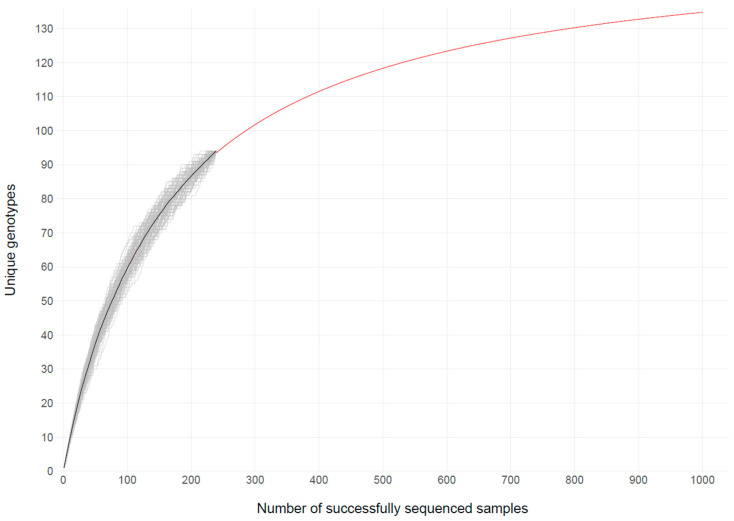
Rarefaction curve. The number of unique identified genotypes was plotted against the number of sequenced samples. Here, 100 collector curves are shown in gray, the average distribution is shown in black, and the extrapolation to estimate the number of genotypes is shown in red.

**Table 1 pathogens-13-00390-t001:** All genotypes are listed by year, as stool and CSF samples were also collected outside the selected respiratory seasons. Genotypes are arranged by species and alphabetically within a species. The most common genotypes are shown in bold.

Species	Genotype *	2019/2020	2021/2022	Total
Enterovirus A	CVA4	0	1	1
	**CVA6**	**3**	**10**	**13**
	CVA10	1	6	7
	EV-A71	2	0	2
Enterovirus B	CVA9	0	2	2
	CVB1	0	1	1
	CVB2	1	2	3
	**CVB3**	**1**	**7**	**8**
	CVB4	1	3	4
	**CVB5**	**5**	**3**	**8**
	E6	1	2	3
	E7	3	0	3
	E11	0	1	1
	E18	0	2	2
	E20	2	0	2
	E21	1	0	1
	E25	1	1	2
	E30	1	0	1
Enterovirus D	**EV** **-** **D68**	**1**	**2**	**3**
Rhinovirus A	RV-A1	6	0	6
	RV-A2	1	1	2
	RV-A8	1	0	1
	RV-A9	1	0	1
	RV-A11	3	1	4
	RV-A12	2	1	3
	RV-A15	1	3	4
	RV-A20	2	2	4
	RV-A21	1	0	1
	RV-A22	1	0	1
	**RV-A24**	**1**	**8**	**9**
	RV-A25	1	3	4
	RV-A28	1	0	1
	RV-A29	0	1	1
	RV-A30	0	1	1
	RV-A31	1	4	5
	RV-A32	1	0	1
	RV-A33	1	0	1
	RV-A34	4	1	5
	RV-A36	1	3	4
	RV-A39	1	0	1
	RV-A46	1	3	4
	**RV-A47**	**2**	**7**	**9**
	RV-A49	1	0	1
	RV-A53	1	3	4
	RV-A55	1	0	1
	RV-A56	1	0	1
	RV-A57	1	1	2
	RV-A58	0	3	3
	RV-A59	0	3	3
	RV-A60	1	0	1
	RV-A61	0	2	2
	RV-A65	0	1	1
	RV-A66	1	0	1
	RV-A67	1	0	1
	RV-A71	1	0	1
	RV-A78	1	2	3
	RV-A80	1	1	2
	RV-A82	1	0	1
	RV-A85	0	2	2
	RV-A100	1	0	1
Rhinovirus B	**RV-B3**	**7**	**1**	**8**
	RV-B4	1	0	1
	RV-B6	2	0	2
	RV-B14	5	0	5
	RV-B26	1	0	1
	RV-B27	2	2	4
	RV-B35	1	0	1
	RV-B42	0	1	1
	RV-B48	1	0	1
	RV-B69	1	2	3
	RV-B70	3	0	3
	RV-B72	1	1	2
	RV-B83	0	2	2
	RV-B84	1	0	1
	RV-B86	2	0	2
	RV-B91	2	0	2
	NA	1	0	1
Rhinovirus C	**RV-C1**	**1**	**4**	**5**
	RV-C5	0	1	1
	RV-C7	1	0	1
	RV-C11	1	2	3
	RV-C12	1	0	1
	RV-C15	2	0	2
	RV-C17	1	0	1
	RV-C24	1	0	0
	RV-C25	0	1	1
	RV-C26	1	0	1
	RV-C33	1	0	1
	RV-C41	2	0	2
	RV-C42	1	1	2
	RV-C43	2	0	2
	RV-C45	1	1	2
	RV-C51	0	1	1
	RV-C53	1	0	1
	RV-C56	0	1	1

* Abbreviations are used according to [30,34].

## Data Availability

Sequences are available from the Swiss Pathogen Surveillance Platform (https://spsp.ch) upon request [39].

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
