# Peer review of "Retrospective Genotyping of Enteroviruses Using a Diagnostic Nanopore Sequencing Workflow"

_pathogens, 2024, doi:10.3390/pathogens13050390_

Round 1
Reviewer 1 Report
Comments and Suggestions for Authors
The manuscript by van Ackeren describes the diversity of enterovirus and rhinovirus species in clinical samples collected in years 2019-2022. The writers adapted the already published and used VP1 sequencing with primers from Nix et al. to a workflow using the nanopore Flongles. The work has been done very well and manuscript is well written. There are, however, some things that can be improved.
The names and abbreviations used for enteroviruses could be according to Simmonds et al. Recommendations for the nomenclature of enteroviruses and rhinoviruses. Archives of Virology (2020) 165:793–797. https://doi.org/10.1007/s00705-019-04520-6. There are no big differences, species names: Enterovirus A (not Enterovirus_A), Rhinovirus A (not Rhinovirus_A); genotypes: coxsackieviruses like CVA9 (not CV_A9), echoviruses like E11 (not E_11), and numbered entroviruses like EV-A71 (not EV_A71). This applies, of course, also to the supplementary material.
Detailed comments are below:
Row 91: “..and 0.2 ng/μL product of the second PCR.” Does here the 0.2 ng/µl product mean that there should be added 0.2ng of the product or should the product be diluted to this given concentration and the 1µl of this product be added. This was a bit confusing.
Beginning from row 154: The total number of samples. First it seemed from Supplementary table 1 that there were 412 samples in total. However, I realized that this was not the case, since the number was the study number. This was a bit confusing.
Rows 157-159: it would be good to mention which samples had a double infection. This was really hard to find from Table S1. Also, quite many samples had Ct values for both entero and rhino PCR? From these it was impossible to find the double infections. I was finally able to find two of the double infections, but this was difficult. Furthermore, many samples had no Ct values for either virus, but these are not discussed in the results or discussion. I think these should be mentioned or discussed or both. For example, it there were no Ct for neither virus, why was it still sequenced? Moreover, some thoughts why they still gave a genotype.
Rows 196-199: The finding of the still unclassified rhinovirus. Possibly sample 32 in Table S1 (?), it could be added to the table that this is still unclassified.
Rows 202-203: Which samples were co-infected with norovirus or bocavirus?
Discussion, beginning from row 258: The part where the SARS-CoV-2 pandemic and its effects are discussed. There is no mention or thoughts if the massive numbers of SARS-CoV-2 diagnostics decreased not only the amount of enterovirus infections, but also the numbers of the diagnostics done for other viruses. As there are no antivirals against entero- or rhinoviruses, the diagnostic capacity was directed to controlling the pandemic and thus SARS-CoV-2 diagnostics.
Reviewer 2 Report
Comments and Suggestions for Authors
van Ackeren et al. describe the use of a rapid tool for enteroviruses genotyping and the distribution of viral genotypes in Zurich, before and during COVID-19 pandemic. Overall it is well written but a revision is required and the discussion section should be enriched.
My comments:
- Page 1, line 40: a brief introduction to enteroviruses cases in Europe would be appreciated.
- Page 2, line 47, “ Residual routine patient samples with previously confirmed enterovirus infection”: how was confirmation performed? Are Real-Time PCR Ct values available?
- Page 2, line 72, “ three consecutive PCRs were performed”: a figure or a table resuming the three PCRs procedures would be appreciated.
- Page 4, figure 1: missing data.
- Page 5, lines 160-166: the description of these results should be rephrased or the representation in figure 3 should be changed. In my opinion it is currently a bit confusing.
- Page 6, lines 176-182: did the authors detect any specific genotype by type of material (CSF, stool, respiratory)?
- Page 9, lines 206-215: a simple statistical analysis on prevalence data might be performed.
- Page 10, lines 222-223: please add more details.
- Page 11, lines 237-238: what would be the advantages of using the authors method for enteroviruses genotyping as compared to the others currently available? Please discuss it.
- Page 11, lines 256-257: possible explanations? Please discuss it.
Comments on the Quality of English LanguageVerbs in both present and past forms are used in the text. Please revise it.
The use of upper and lower cases for viruses names should be revised too.
Round 2
Reviewer 2 Report
Comments and Suggestions for Authors
All raised points have been addressed.